# MEASURING HUMAN-CLIP ALIGNMENT AT DIFFERENT ABSTRACTION LEVELS

**Pablo Hernández-Cámara, Jorge Vila-Tomás, Jesús Malo & Valero Laparra**
Image Processing Lab
University of Valencia
Valencia, Spain
`{pablo.hernandez-camara, jorge.vila-tomas, jesus.malo, valero.laparra}@uv.es`

## ABSTRACT

Measuring the human alignment of trained models is gaining traction because it is not clear to which extent artificial image representations are proper models of the visual brain. Employing the CLIP model and some of its variants as a case study, we showcase the importance of using different abstraction levels in the experiments, because when measuring image distances, the differences between them can have lower or higher abstraction. This allows us to extract richer conclusions about the models while showing some interesting phenomena arising when analyzing the models in a depth-wise fashion. The conclusions extracted from our analysis identify the size of the patches in which the image is divided as the most important factor in achieving a high human alignment for all the abstraction levels. We found that the method used to compute the model distances is crucial to avoid alignment drops. Moreover, replacing the usual softmax activation with a sigmoid also increases the human alignment at all the abstractions especially in the last model layers. Surprisingly, training the model with Chinese captions or medical data gives more human-aligned models but only on low abstraction levels.

## 1 INTRODUCTION

Measuring the alignment of the human brain and artificial networks can lead to models with better generalization abilities, more robust to adversarial attacks, better cross-domain learning (Nanda et al., 2021; Aho et al., 2022; Fel et al., 2022; Sucholutsky & Griffiths, 2023; Muttenthaler et al., 2023), and it can help neuroscientists to study the underlying principles of human behaviour (Brette, 2019; Martinez-Garcia et al., 2019; Allen et al., 2022; Conwell et al., 2022; Caucheteux & King, 2022; Hernández-Cámara et al., 2023). The alignment with human behaviour can be analyzed at different complexities or abstraction levels. For example, when measuring the alignment with the way humans measure the distance between images, the distances can have very different scales because of the complexity of the images and/or the distortions. It is not the same measuring the distances between an image and the same image with a small quantity of noise than between images from different classes as shown in Figure 1. Measuring alignment at only one level of abstraction and only in the model deepest layer may miss interesting behaviours that occur at other abstraction levels and/or in other model layers.

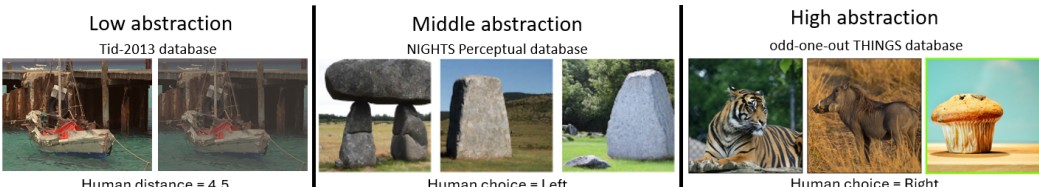

Figure 1: Example of the different abstractions when measuring image distances. **Low abstraction:** measuring distances between pairs of images where one is the reference and the other is a distorted version with a small noise (TID2013). **Middle abstraction:** triplet preferences between images of the same class but with semantic differences, such as the number of objects (NIGHTS perceptual). **High abstraction:** triplet preferences between objects of different classes (odd-one-out THINGS).

Table 1: **Our method**: we isolated each factor to analyze it independently. Within the big conceptual categories, *model design* and *training procedure*, each column shows the different factors that we varied in our study and each row shows the different options explored.

| MODEL DESIGN | | TRAINING PROCEDURE | |
|---|---|---|---|
| Architecture | Last activation funct. | Languages | Data type |
| base-patch16 base-patch32 large-patch14 large-patch14-336 | Softmax | English | Natural images |
| base-patch16 | Softmax Sigmoid | English | Natural images |
| base-patch16 | Softmax | English Chinese | Natural images |
| base-patch16 | Softmax | English | Natural images Medical images |

Although there have been some works studying the human alignment of multimodal models, especially CLIP (Geirhos et al., 2021; Muttenthaler et al., 2022; 2023), here we propose to analyze in detail the way some training procedures and model design details affect its human alignment. Interestingly, we do not only measure the alignment in a single abstraction level as the majority of works do, but we measure at different abstraction levels: from measuring distances between images with small noise distortions to intra-class distances between images with different semantic meanings and finishing with inter-class image distances. Moreover, we also analyze how the alignment changes depending on the layer depth and training/architecture factors.

## 2 METHODS

Following Sucholutsky et al. (2023), we measure the alignment between the human brain and different neural networks, using behavioural data about image distances. We consider not only a single dataset but three different data sources of different abstraction levels. We measure how human alignment at these three abstraction levels changes layer by layer depending on different model design choices or training procedures. To do that, we successively isolate the different factors, fixing all but one, as shown in Table 1. Each row describes one experiment, and each column lists the explored options. Analyzing all the possible combinations is not possible because they are not available.

### 2.1 DATA: ABSTRACTION LEVELS IN DISTANCE MEASUREMENTS

We use behavioural data about image similarity consisting of human-assessed distances between pairs of images, or human preference in the triplets scenario. We use data from three different abstraction levels of *distortion*, where the differences between the original image and the *distorted* versions span from lower to higher abstraction. We call the different levels: low, middle, and high. Figure 1 shows an example from each of the abstraction levels considered.

For the low abstraction, we use a perceptual quality database, TID-2013 (Ponomarenko et al., 2015). It consists of pairs of images where one is a reference image and the other is a noisy version of it, and the mean opinion score regarding how much humans see the difference between the two images. In the middle abstraction, we use the NIGHTS perceptual data (Fu et al., 2023). It consists of intra-class synthetic image triplets where the differences between the images cover a wide range of variations that include semantic differences, such as object layouts, poses or quantities. We use the THINGS similarity dataset (Hebart et al., 2020) for the high abstraction. It is a subset of the whole THINGS dataset, comprising 4.70 million triplet odd-one-out similarity judgments for more than 1800 classes. It consists of triplets of inter-class images and therefore the differences are much more abstract. Appendix A shows more details and examples of the different datasets.

### 2.2 MODELS

We restrict ourselves to models trained by third-party institutions to avoid dependences on training procedures. We focus on evaluating CLIP Radford et al. (2021) model, trained to predict correspondences between images and their captions, and analyze how some model design variations or training procedures affect its human alignment at the three different abstractions. For the different

model architectures, we analyze the *base-patch16*, *base-patch32*, *large-patch14*, and *large-patch14-336* versions. We compare the effect of the final activation function comparing CLIP and SigLIP (Zhai et al., 2023), which replaces the usual softmax activation function with a sigmoid. We analyze if the language used in the training has some effect on the alignment by comparing CLIP (trained in English) with Chinese-CLIP (Yang et al., 2022) (trained in Chinese). Finally, we analyze the effect of the training data by comparing CLIP (trained with natural images) with BiomedCLIP (Zhang et al., 2023) (trained with medical images and texts). Appendix B shows more details of the models.

As a baseline, we used some low abstraction image quality assessment (IQA) models: the classical SSIM (Wang et al., 2004), and more recent models based on neural networks such as LPIPS (Zhang et al., 2018) and PerceptNet (Hepburn et al., 2020), which has biological inspiration.

## 3 EXPERIMENTS AND RESULTS

Following the methodology stated in table 1, we fixed all the factors but one in each experiment. We represent models that appear in more than one experiment always with the same color for easier comparison. All the measures have been done in a depth-wise fashion to facilitate the analysis in a more granular way, so all the figures show the model's depth at the x-axis. As some models have a different number of layers, to plot them all together the x-axis represents the percentage of the network (0% being the image space and 100% being the deepest layer - projection). We compute distances between images using the Euclidean metric normalizing the differences so that all features have unit mean (see appendix C for more details about the distance measurement procedure and its crucial relevance). Regarding the alignment measurement, the y-axis shows Spearman correlation between model distances and mean opinion score in the low-abstract level and accuracy between model distance-based preference and human triplet preference in the mid/high-abstract levels.

### 3.1 DIFFERENT ARCHITECTURE

First, we analyze how the alignment changes depending on the CLIP model architecture: number of layers (*base* vs *large*), patch size (*base-patch16* vs *base-patch32*) and number of patches (*large-patch14* vs *large-patch14-336*). Figure 2 shows that the *base-patch32* variant gets the best result in the three abstraction levels but especially in the low abstract data. This shows that *the patch size is the most important factor to consider to attain high human alignment*.

Although the differences between the other three model sizes are small, the large model with more patches *large-patch14-336* achieves worse results across the three abstraction levels. Also, the more aligned model, *base-patch32*, shows the maximum human alignment around the 70% of the model depth for the mid and high abstract datasets showing an alignment decrease in the final layers before an increase in the projection. It shows a much more flat relation with layer depth in the TID2013 dataset before a final drop in the projection. Note that all the models consistently outperform all IQA baselines in the high abstract data and their middle layers also in the middle abstract. They are expected to perform worse than PerceptNet on TID-2013 because it was trained specifically on that abstraction level.

### 3.2 DIFFERENT ACTIVATION FUNCTION

Second, we analyze how the alignment changes depending on the CLIP last activation function. Figure 3 shows that the SigLIP model gets higher human alignment than CLIP in the low-abstract TID2013 although their projections get closer. In the mid-abstraction level, the CLIP model is more

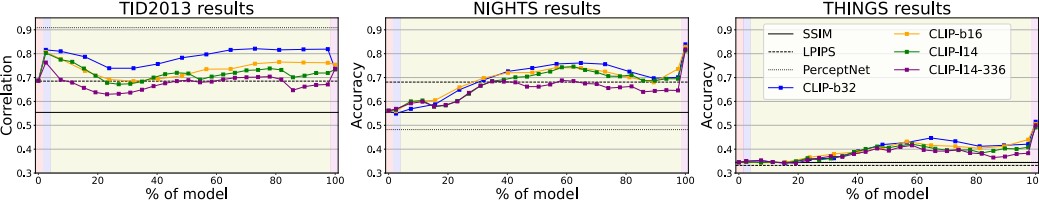

Figure 2: Human alignment with TID2013 (left), NIGHTS (center) and THINGS (right) analyzed layer-by-layer depending on the model size.

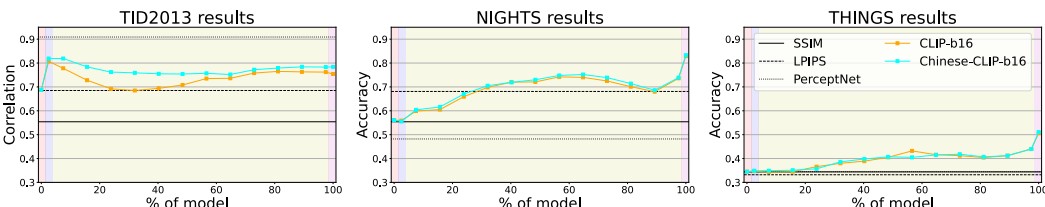

Figure 3: Human alignment with TID2013 (left), NIGHTS (center) and THINGS (right) analyzed layer-by-layer depending on the activation function.

aligned in the early layers but less in the deeper layers. In the high-abstract data, both models perform similarly until the deepest layers, where the CLIP model surpasses the SigLIP. Moreover, it is interesting to notice that the SigLIP model shows a decrease of alignment in its first layer (embeddings) especially in the middle-level data but it does not happen in the low-level data, where it shows a big alignment increase. Both models outperform IQA baselines in the three abstract data, except for PerceptNet in the low-abstract scenario.

## 3.3 DIFFERENT LANGUAGE

Third, we analyze how the alignment changes depending on the text caption language. Surprisingly, Figure 4 shows that there are differences in the models when checked at the lowest abstraction, especially at early layers where CLIP with Chinese captions results in a more human-aligned model. However, these differences completely disappear when the models are tested at mid or high-abstraction levels. As before, it is interesting to note that both models outperform the SSIM and LPIPS across the three abstraction levels especially when measuring at their intermediate/final layers but not at the beginning of the model in the mid-abstract level.

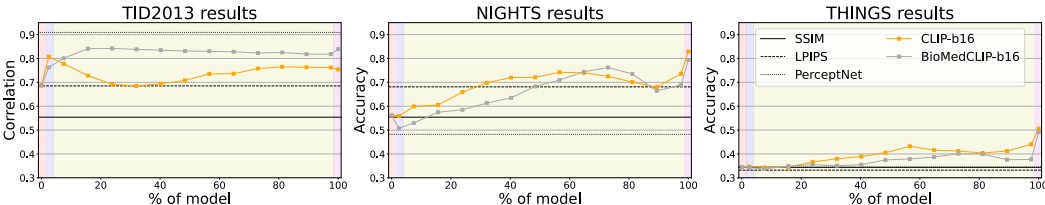

Figure 4: Human alignment with TID2013 (left), NIGHTS (center), and THINGS (right) analyzed layer-by-layer depending on the language used in the training.

## 3.4 DIFFERENT TRAINING DATA

Finally, we analyze how human alignment changes depending on the training data type. In Figure 5 we found that the CLIP model trained with medical data (med-CLIP) is much more human-aligned than the CLIP trained with natural data (nat-CLIP) in the low abstraction level scenario. This difference in alignment is reduced when tested in more abstract data. However, in the mid and high abstractions, nat-CLIP is more aligned than med-CLIP in the early layers, but in the final layers, med-CLIP again surpasses the model trained with natural data. When comparing against the baselines, med-CLIP performs better than both SSIM and LPIPS at the lower and higher levels, but both models are closely matched to LPIPS at the mid and high abstraction levels.

Figure 5: Human alignment with TID2013 (left), NIGHTS (center) and THINGS (right) analyzed layer-by-layer depending on the data used in the training.

## 4    DISCUSSION & CONCLUSSION

Here we hypothesize some possible causes for the results described above. First, we found that the relative size of the image and the patches used in its analysis is critical for the alignment between humans and machines. For fixed-size images, bigger patches are better (blue curve is the best in Fig.2), while for fixed-size patches, smaller images are better (green curve is better than purple curve in Fig. 2). Both effects are consistent: when the scene is partitioned in too-small regions the machine-human alignment is worse. We hypothesize there are optimal scales for scene analysis (see Appendix D Lowe (2004); Lazebnik et al. (2006)): when using the wrong scale spatial relations between regions are harder to describe and hence the behavior of the model maybe less human.

Another interesting result appears when no-normalization of the features over the datasets is considered before distance computation (Fig. 9 in Appendix C): there is a systematic alignment drop at mid-depth in all the models. This may make qualitative sense: the meaning of the visual features along the architecture should be close to low-level visual primitives in early layers, and be more abstract (closer to a conceptual description) in late layers where training considers the distance with the textual representation. Given this evolution, distances at different layers would be aligned with human (visual) opinion early in depth, but the relation may be obscured later in the architecture leading to a drop in the alignment. Preliminary inspection of the responses suggests that this behavior comes from the fact that a single feature takes most of the energy of the signal from around 50% depth. That singularity was the reason to propose difference normalization over the datasets in our experiments. This normalization-over-images has two interesting consequences: (1) the drop in alignment disappears, and, (2) the increase in alignment happens in later layers for higher abstractions for almost all the models. As shown in Fig 2, for low-abstraction (left) a substantial increase in alignment happens as early as at 10% depth. However, for middle abstraction (center), the substantial increase happens about 30% depth, and for high abstraction (right) alignment remains small up to 50% depth. This is consistent with previously observed behaviour going from low-level to higher-level problems both in CNN and tranformers (Zeiler & Fergus, 2014; Ghiasi et al., 2022).

Regarding the activation function, a hypothesis for SigLIP's superior alignment over base CLIP in Fig. 3 is that the CLIP softmax enforces high output values to highlight the correct answer. Whereas, in the SigLIP sigmoid high values would saturate the non-linearity, thus the model will enforce outputs in the sigmoid range. Having a wide output range is problematic with out-of-distribution data due to a more unpredictable output scale. Additionally, SigLIP's authors state that the sigmoid stabilizes the training (Zhai et al., 2023), which could lead to a better performance model, making it harder to directly relate the difference only to the different activation functions.

The differences in human alignment between the CLIP models trained in different languages could be the result of language differences, however it could also be due to the different datasets, training procedures, pretraining, or tokenizers between the models. As opposed to the experiments on region sizes where we compare models with uniform training procedures from the same organization (OpenAI), this experiment uses models from different organizations. Therefore it is difficult to make conclusions on the effect of the language. That is a clear limitation of this experiment.

Finally, regarding the nature of the training data, we found more differences in the low abstraction case. We hypothesize that medical images contain finer details (high-frequency) than natural images. Thus a model trained on medical images is more aligned with TID2013 where the distortions are different types of small-amplitude noises. Many medical models have historically used high-frequency noise textures to classify images (Castellano et al., 2004; Jaén-Lorites et al., 2022).

In conclusion, analyzing CLIP's human alignment for different abstraction level problems and at different model depths we found (1) a global drop in machine-human similarity for higher abstraction problems, and (2) after response normalization, the layer with biggest similarity shifts towards higher depths for higher abstraction levels. This highlights the relevance of a depth-wise analysis, suggest that deeper layers indeed correspond to higher abstraction, but also reveal the progressive departure between human and CLIP representations. (3) We found that for all the abstractions, the right balance between image and patch size is much more important than having more layers or more patches. Finally, we found that training the model with medical images instead of natural images leads to a much more human-aligned model in the low abstract scenario but, as happened with the language, it does not have clear effect at higher abstractions. Moreover, we showed that many of the analyzed models outperform state-of-the-art image quality models, such as SSIM or LPIPS.

ACKNOWLEDGMENTS

This work was supported in part by MICIIN/FEDER/UE under Grant PID2020-118071GB-I00 and PDC2021-121522-C21, in part by Spanish MIU under Grant FPU21/02256, in part by Generalitat Valenciana under Projects GV/2021/074, CIPROM/2021/056 and CIAPOT/2021/9 and in part by valgrAI - GVA. The authors gratefully acknowledge the computer resources at Artemisa and the technical support provided by the Instituto de Fisica Corpuscular, IFIC(CSIC-UV). Artemisa is co-funded by the European Union through the 2014-2020 ERDF Operative Programme of Comunitat Valenciana, project IDIFEDER/2018/048.

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

## A  DATASETS

We used three datasets of different abstraction levels. More particularly, we used TID-2013, NIGHTS perceptual and odd-one-out THINGS for the low, middle and high abstraction levels.

First, the TID-2013 database consists of 3000 image pairs plus the Mean Opinion Score (MOS), which is a continuous value that indicates how much humans see the difference between a given pair. By definition, the MOS increases as humans see less the differences. Each pair is composed

of an original image and a distorted version of it by a small noise (distortion) as some examples show in Figure 6. To compute the human alignment at this abstraction level we pass each original and distorted image through the model. Then we compute a model distance between each database pair and finally, we calculate the Spearman correlation between model distances and MOS to get an alignment value.

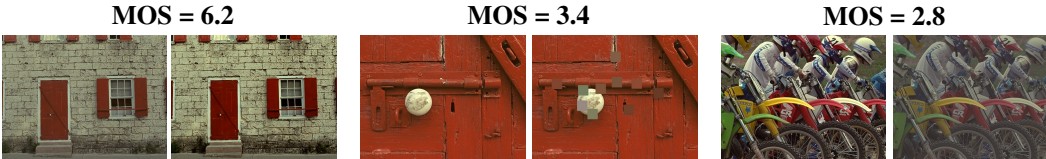

Figure 6: Example of three different TID-2013 pairs. For each pair, the left image is the original one and the right image is the distorted version and they include the corresponding Mean Opinion Score.

Second, the NIGHTS perceptual is a database of human similarity judgments over image pairs that are alike in diverse ways such as object poses, layouts and numbers. It is composed of 20019 image triplets with human scores of perceptual similarity. Each triplet consists of a reference image and two distortions plus the human reference that indicates which distorted version humans found closer to the original one, as shown in Figure 7.

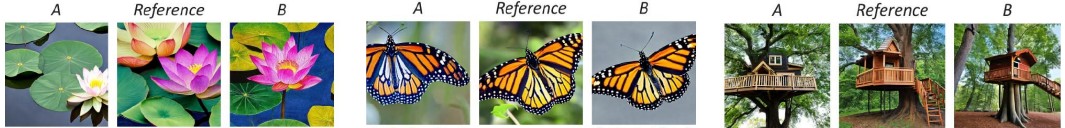

Figure 7: Example of three different NIGHTS triplets. Each triplet has the original image in the middle and two distorted versions A and B. In the three triplets, human preference is left, meaning that humans found B distorted closer to the original one than A closer to the original.

Finally, the odd-one-out THINGS dataset consists of more than 4.7 million inter-class similarity triplets from more than 1800 classes. For each of the triplets, humans indicate which of the images is the most dissimilar to the other two, so that they have to choose between three options, as shown in Figure 8.

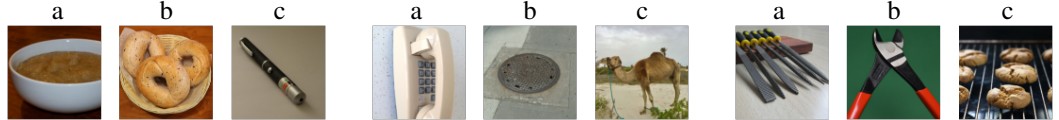

Figure 8: Example of three different THINGS odd-one-out triplets. Each triplet has three different images (a, b and c) from different classes. In the three triplets, human preference is c, meaning that humans found the c image to be the odd-one-out.

## B  MODELS

As stated in the text, we restricted ourselves to open models trained by third-party institutions to avoid dependence on training procedures. Therefore, not all the model factor combinations are available for its study. We focus on analyzing CLIP model (Radford et al., 2021) and different training procedures and architecture design variations. Table 2 shows the names and specific versions of the different analyzed models. We took all the models from Hugging Face Model Zoo[1].

To compare different CLIP architectures we used four versions: *base-16*, *base-32*, *large-14* and *large-14-336*. The base/large indicates if the model has 13 or 25 layers. The number after base/large indicates the pixel size of the patches in which the image is divided when used as input to the model. Therefore, base-16 indicates that the image is divided into $16 \times 16$ pixel patches and that the model has 13 layers. In general, all the models resize the images to $224 \times 224$ pixels before dividing it into patches except for the last model *large-14-336*, which resize the image to $336 \times 336$ pixels and therefore has more patches than *large-14*.

---

[1]https://huggingface.co/models

Table 2: Details of the different analyzed multimodal models. All of them were used from Hugging-Face: `https://huggingface.co/models`.

| MODEL | HuggingFace name |
|---|---|
| CLIP | openai/clip-vit-base-patch16 |
| CLIP | openai/clip-vit-base-patch32 |
| CLIP | openai/clip-vit-large-patch14 |
| CLIP | openai/clip-vit-large-patch14-336 |
| SigLIP | google/siglip-base-patch16-224 |
| Chinese-CLIP | OFA-Sys/chinese-clip-vit-base-patch16 |
| BiomedCLIP | microsoft/BiomedCLIP-PubMedBERT_256-vit_base_patch16_224 |

To compare the different final activation functions, we used CLIP and SigLIP models. The only difference between them is that the SigLIP model replaces its final softmax activation function with a sigmoid function. To compare the training language we compare between CLIP and Chinese-CLIP, trained with Chinese captions instead of English ones. Finally, to compare the type of data we compare CLIP and BiomedCLIP, trained on public medial data instead of natural images. For these three experiments, we used the *base-16* versions, so that they have 13 layers, resize the images to $224 \times 224$ pixels and divide them into $16 \times 16$ pixel patches.

## C  DISTANCE MEASUREMENT

As stated in the text, we used a normalized Euclidean metric to measure distances between image model outputs: we normalize the difference in each feature so that all features have unit mean over a wide class of scenes. However, that is not the only option to compute the distances. Figure 9 shows how alignment depends on the different explored options to compute distances/similarities between images for the CLIP-*base-16*. More particularly, we compare between three distance measurements:

- Euclidean: $||a - b||$
- Normalized Euclidean: $||norm(a - b)||$
- Cosine distance: $1 - \frac{a \cdot b}{||a|| \cdot ||b||}$

Naive use of plain Euclidean distance leads to a drop in the alignment for 50% depth of the models. We obtained that measuring Euclidean with the normalized differences is needed to avoid an alignment drop that if not happens at the middle layers. We found that this drop happens in all the CLIP model sizes except for the *base-32*, the one with the biggest patch size. We did not find differences between using the Euclidean distances or the usual cosine similarity (inner product between vectorized outputs over the norm of the outputs).

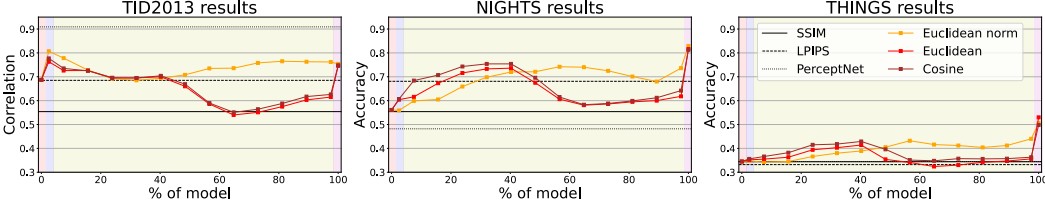

Figure 9: Human alignment with TID2013 (left), NIGHTS (center) and THINGS (right) analyzed layer-by-layer depending on how the distance/similarity between images are computed.

## D  CONJECTURES ABOUT THE BEHAVIOR

Figure 10 illustrates the qualitative reasons for the main findings of the work: (i) the drop of the alignment with human vision at mid-depth when using plain Euclidean metrics and (ii) the relevance of the balance between the sizes of the image and the patches to analyze the image.

These effects may come from: (i) the evolution of the meaning of the features (Fig. 10.A) from low-level visual primitives in early layers to (less-visual) descriptions more suitable to be compared

with textual representations in later layers. When the representation starts to be less-visual it is not surprising that plain Euclidean distance fails to reproduce human behavior. (ii) the balance between the spatial struture of the image and the proper scale to analyze it (Fig. 10.B). Classical hand-crafted methods to extract image descriptors for successful classification used to operate at different scales, and there is a scale-hierarchy that depends on the spatial structure of the image Lowe (2004); Fei-Fei & Perona (2005); Lazebnik et al. (2006). For instance, faces such as the one in Fig. 10.B have certain objects of certain scale (eyes, nose, mouth) in specific spatial relations. At high resolution (analysis regions of small size) objects are hard to identify and spatial relations are hard to establish. This effect may be happening to the transformers in CLIP too. This may explain the critical effect of the patch size in relation to image size.

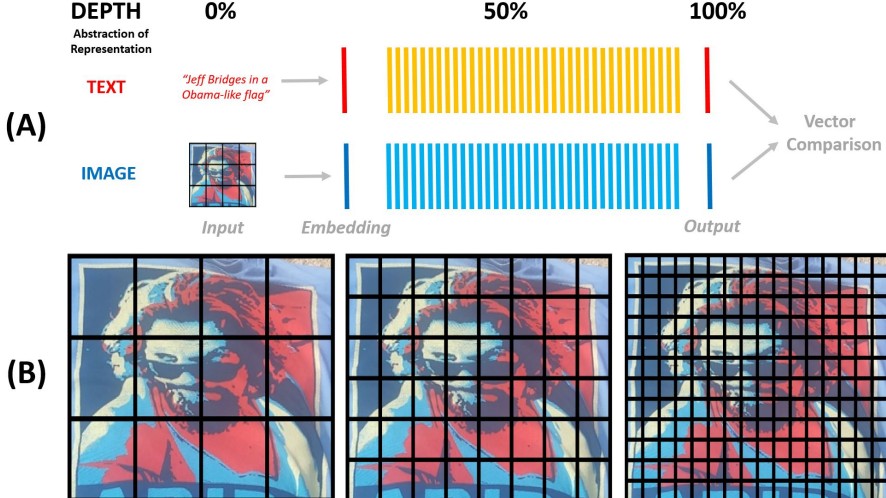

Figure 10: (a) Expected abstraction along the representation: in the image channel low-level image primitives are expected in the embedding and early layers, while more conceptual abstract representations are expected at late layers. (b) The scale of the image regions considered by the models is key in order to properly describe the spatial structure in the images Lowe (2004); Lazebnik et al. (2006).

