# OpenReview forum: "Measuring Human-CLIP Alignment at Different Abstraction Levels"
_ICLR.cc/2024/Workshop/Re-Align — ICLR 2024 Workshop Re-Align Poster_

### Official Review · Reviewer_yz81 · 2024-02-23
**A paper that explores alignment in CLIP models from the perspective of image similarity with a substantial number of experiments on a limited range of models.**

**Rating:** 2
**Fit:** 3
**Confidence:** 2

**Workshop Review:**

Strengths:
- Framing the question of alignment in terms of model and human assessment of image distance is interesting and allows for nice quantitative experiments.
- Given that this is only a 5 page paper, a substantial number of experiments are run, allowing a reasonably thorough exploration.

Weaknesses:
- One of the challenges of studying foundation models such as CLIP is that it is very hard to perform experiments that control for training details. For instance, do the results of an experiment on two CLIP models (independently trained) arise from the training/architecture differences we are aware of or do they arise from nuances of training/data that are not publicly known. Further, we rarely have multiple instances of a model with fixed characteristics, so it is hard to determine what is training noise. This is mostly unavoidable given the compute budgets of most research groups, but it should definitely be cited as a limitation and conclusions should be framed accordingly.
- There are quite a few details that should be spelled out more completely for clarity. For instance, not all readers will be familiar with the different versions of the CLIP architecture. This should be spelled out explicitly, especially since these architectural features are central to some of the experiments. The details around the three datasets that are used for evaluation should also be stated since not all readers will be familiar with these. More example images in the appendix would be useful.

Nitpicks:
- When describing the TID-2013 dataset, it might be useful to say a little more what the ‘opinion score’ is (e.g., what kinds of values does it take, is it continuous or discrete?)
- There is a space missing in the last paragraph of Section 2.1, “… images.In the…”

Questions:
- What is a model’s “inner domain” (first paragraph, page 1)?
- What is the difference between large-patch14 and large-partch14-336?

**Reason For Not Giving Higher Score:**

Given the very limited number of models and the lack of control of their training/data, this reviewer feels that further experiments are needed to have confidence in some of the paper’s claims. Additional proofreading and details would make the paper stronger.

**Reason For Not Giving Lower Score:**

The research direction is interesting and many of the initial findings are a strong foundation for future work.

**Reviewer Domain:**

machine learning

---

> ### Author Response · Authors · 2024-04-25
>
> > One of the challenges of studying foundation models such as CLIP is that it is very hard to perform experiments that control for training details. For instance, do the results of an experiment on two CLIP models (independently trained) arise from the training/architecture differences we are aware of or do they arise from nuances of training/data that are not publicly known. Further, we rarely have multiple instances of a model with fixed characteristics, so it is hard to determine what is training noise. This is mostly unavoidable given the compute budgets of most research groups, but it should definitely be cited as a limitation and conclusions should be framed accordingly.
>
> We agree with the statement and in the camera-ready version, we will cite it as a limitation. Although training our models will be the best option to be sure where the differences come from, it will create our own bias regarding the training procedure/data. Moreover, as the reviewer says, the computational power needed is prohibitive and therefore we decided to use the more similar models as possible.
>
> > There are quite a few details that should be spelled out more completely for clarity. For instance, not all readers will be familiar with the different versions of the CLIP architecture. This should be spelled out explicitly, especially since these architectural features are central to some of the experiments. The details around the three datasets that are used for evaluation should also be stated since not all readers will be familiar with these. More example images in the appendix would be useful. The details around the three datasets that are used for evaluation should also be stated since not all readers will be familiar with these. More example images in the appendix would be useful.
>
> In the camera-ready version, we extended Appendix B Models to explain in detail the differences between all the analyzed models. We also added a new Appendix A Datasets with more examples and details about the different data used at each abstraction level.
>
> > When describing the TID-2013 dataset, it might be useful to say a little more what the ‘opinion score’ is (e.g., what kinds of values does it take, is it continuous or discrete?)
>
> We described it in more detail in the new appendix A Datasets.
>
> > There is a space missing in the last paragraph of Section 2.1, “… images.In the…”
>
> Thanks, changed.
>
> > What is a model’s “inner domain” (first paragraph, page 1)?
>
> We wanted to say the model’s inner representation, i.e. the outputs of the final/deepest model layer. To be more precise, we changed it to “model’s deepest layer”.
>
> > What is the difference between large-patch14 and large-partch14-336?
>
> The only difference is the number of patches: CLIP models usually resize images to 224x224, while the large-pathc14-336 resizes it to 336x336. Both models have a patch size of 14 so the only difference is the overall number of patches.

---

### Official Review · Reviewer_qXb8 · 2024-02-26
**Interesting new evaluation method and results, minor clarity issue**

**Rating:** 2
**Fit:** 3
**Confidence:** 3

**Workshop Review:**

Novelty/interest to the community:
I believe that the idea of systematically evaluating models at different levels of abstraction will be of great interest to the realign workshop community. The approach taken by the authors is novel and they have an interesting finding that larger patch sizes have greater human alignment at all abstraction levels.

Clarity:
Overall the paper was written clearly and the motivation, experiments, and conclusions were easy to follow. I did have trouble understanding exactly how the authors were getting outputs from their clip model that they could then compute correlation scores on. For example, the task for the first level of abstraction involves predicting the perceptual similarity of two images, but the authors did not explicitly state how they computed perceptual similarity using clip embeddings for each image. I assume they are computing cosine similarity, but I recommend the authors include the full procedure for making a judgement using clip for each level of abstraction task in the appendix.

Correctness:
The experiments and analysis appear to be correct.

**Reason For Not Giving Higher Score:**

There were some minor clarity issues in explaining the full experimental procedure.

**Reason For Not Giving Lower Score:**

The work is well motivated. The evaluation approach and results will be of interest to the community.

**Reviewer Domain:**

machine learning

---

> ### Author Response · Authors · 2024-04-25
>
> > Overall the paper was written clearly and the motivation, experiments, and conclusions were easy to follow. I did have trouble understanding exactly how the authors were getting outputs from their clip model that they could then compute correlation scores on. For example, the task for the first level of abstraction involves predicting the perceptual similarity of two images, but the authors did not explicitly state how they computed perceptual similarity using clip embeddings for each image. I assume they are computing cosine similarity, but I recommend the authors include the full procedure for making a judgement using clip for each level of abstraction task in the appendix.
>
> For the camera-ready version, we have included information about how we measure the model distances and human alignment at each abstraction level at the beginning of section 3. Experiments. Also, we have included an appendix about different distance measurement procedures and how they affect the alignment.

---

### Official Review · Reviewer_cWfC · 2024-02-27
**Measuring abstraction levels is interesting**

**Rating:** 2
**Fit:** 3
**Confidence:** 1

**Workshop Review:**

### Strengths
- Comparing human alignment between models across levels of abstraction and across layers is compelling and of interest to the community.
- The authors construct reasonable experiments that are potentially a good fit for a workshop.
- The tentative conclusions are prima facie plausible.


### Weaknesses
- The differences in human alignment between the CLIP models trained in different languages could be the result of language differences, however it could also be due to the different datasets, training procedures, pretraining, or tokenizers between the models. As it stands, these are not controlled for experimentally and it is therefore difficult to make conclusions.
- Similarly, the differences in accuracy between CLIP and SigCLIP (and medical images vs CLIP) make it hard to say whether the human alignment differences come from the activation function or the training recipe.
- Should we be worried about confounding inter vs intra class effects? The middle abstraction level consists of intra-class images only, the higher level of abstraction inter-class.
- The LPIPS and SSIM baselines are the same across the TID2013 and NIGHTS datasets. This is probably a mistake?

### Misc
- Fig. 1 has labeled low/middle/high complexity. Should this be low/middle/high abstraction?
- The CLIP b-16 and l-* models experience sharp decreases in alignment correlation at similar middle layers for the low-abstraction dataset, and then increase. Is this due to a resizing of the spatial activations?

**Reason For Not Giving Higher Score:**

I was between a 1 and a 2. While I think there are correctness/experimental issues, the paper is novel and presents interesting ideas.

**Reason For Not Giving Lower Score:**

See above.

**Reviewer Domain:**

machine learning

---

> ### Author Response · Authors · 2024-04-25
>
> > The differences in human alignment between the CLIP models trained in different languages could be the result of language differences, however it could also be due to the different datasets, training procedures, pretraining, or tokenizers between the models. As it stands, these are not controlled for experimentally and it is therefore difficult to make conclusions. Similarly, the differences in accuracy between CLIP and SigCLIP (and medical images vs CLIP) make it hard to say whether the human alignment differences come from the activation function or the training recipe.
>
> We agree with the statement and in the camera-ready version, we will cite it as a limitation. Only the different size analysis uses models from the same organization (OpenAI) while the rest of the comparisons analyze models from different organizations and therefore follow different training receipts. Although training our models will be the best option to be sure where the differences come from, it will create our own bias regarding the training procedure/data. Moreover, as reviewer 3 says, the computational power needed is prohibitive and therefore we decided to use the more similar models as possible.
>
> > Should we be worried about confounding inter vs intra class effects? The middle abstraction level consists of intra-class images only, the higher level of abstraction inter-class.
>
> We are not really sure about what the reviewer means with this question. On the one hand, the middle abstraction level consists of intra-class images where the differences include different image layouts, poses and number of objects. On the other hand, the high abstract level consists of inter-class image triplets, where each image is from a different class and therefore the differences between them are of higher level than in the middle abstraction level.
>
> > The LPIPS and SSIM baselines are the same across the TID2013 and NIGHTS datasets. This is probably a mistake?
>
> No, the values are correct. Notice that although they are similar values (but not the same), on TID2013 they represent correlations while on NIGHTS they are accuracies.
>
> > Fig. 1 has labeled low/middle/high complexity. Should this be low/middle/high abstraction?
>
> Yes, thanks, we have changed it to be more consistent in the camera-ready version.
>
> > The CLIP b-16 and l-* models experience sharp decreases in alignment correlation at similar middle layers for the low-abstraction dataset, and then increase. Is this due to a resizing of the spatial activations?
>
> No, there is no resizing of spatial activations in any model. We now know that the sharp alignment drop is due to the value explosion of one feature and that if we normalize the output so that all the features have the same mean this decrease disappears. The new appendix C Distance Mesarumenets explains it in more detail. We are still working on understanding why this feature suddenly increases its value from a given layer.

---

### Decision · Program_Chairs · 2024-03-02

Accept (Poster)